# Comparison of Disk Diffusion, E-Test, and Broth Microdilution Methods for Testing In Vitro Activity of Cefiderocol in *Acinetobacter baumannii*

**DOI:** 10.3390/antibiotics12071212

**Published:** 2023-07-20

**Authors:** Natalia Kolesnik-Goldmann, Helena M. B. Seth-Smith, Klara Haldimann, Frank Imkamp, Tim Roloff, Reinhard Zbinden, Sven N. Hobbie, Adrian Egli, Stefano Mancini

**Affiliations:** Institute of Medical Microbiology, University Zurich, Gloriastrasse 28/30, 8006 Zurich, Switzerland; nkolesnik@imm.uzh.ch (N.K.-G.); hsethsmith@imm.uzh.ch (H.M.B.S.-S.); khaldimann@imm.uzh.ch (K.H.); imkamp@imm.uzh.ch (F.I.); troloff@imm.uzh.ch (T.R.); xzbe@zhaw.ch (R.Z.); sven.hobbie@imm.uzh.ch (S.N.H.); aegli@imm.uzh.ch (A.E.)

**Keywords:** cefiderocol, *A. baumannii*, disk diffusion, broth microdilution, E-test, iron-depletion

## Abstract

The reference method for cefiderocol antimicrobial susceptibility testing is broth microdilution (BMD) with iron-depleted-Mueller-Hinton (ID-MH) medium, whereas breakpoints recommended for disk diffusion (DD) are based on MH-agar plates. We aimed to compare the performance of the commercial BMD tests ComASP (Liofilchem) and UMIC (Bruker), and DD and E-test using MH- and ID-MH-agar plates with the reference BMD method using 100 carbapenem-resistant-*A. baumannii* isolates. Standard BMD was performed according to the EUCAST guidelines; DD and E-test were carried out using two commercial MH-agar plates (BioMérieux and Liofilchem) and an in-house ID-MH-agar plate, while ComASP and UMIC were performed according to the manufacturer’s guidelines. DD performed with the ID-MH-agar plates led to a higher categorical agreement (CA, 95.1%) with standard BMD and fewer categorization errors compared to the commercial MH-agar plates (CA BioMérieux 91.1%, Liofilchem 89.2%). E-test on ID-MH-agar plates exhibited a significantly higher essential agreement (EA, 75%) with standard BMD compared to the two MH-agar plates (EA BioMérieux 57%, Liofilchem 44%), and showed a higher performance in detecting high-level resistance than ComASP and UMIC (mean log2 difference with standard BMD for resistant isolates of 0.5, 2.83, and 2.08, respectively). In conclusion, DD and E-test on ID-MH-agar plates exhibit a higher diagnostic performance than on MH-agar plates and the commercial BMD methods. Therefore, we recommend using ID-MH-agar plates for cefiderocol susceptibility testing of *A. baumannii*.

## 1. Introduction

Cefiderocol is often the last active agent in gram-negative bacteria before pan-resistance ensues, in particular for carbapenem-resistant *Acinetobacter baumannii* (CRAB) [1], which represents a major cause of hospital-acquired infections worldwide [2]. Cefiderocol exploits its siderophore moiety to gain access into bacterial cells through active iron transporters. However, this uniqueness also poses a great challenge for antibiotic susceptibility testing (AST), which is reflected by differences in the interpretative criteria established by the EUCAST, CLSI [3,4], and FDA [5]. In fact, accurate in vitro AST requires iron-depleted conditions (ID-MH) to induce siderophore-mediated entry. Despite the complexity of the method and the difficulties that sometimes occur with reading the minimal inhibitory concentrations (MICs) due to the emergence of trailing points, there is a consensus that broth microdilution (BMD) represents the gold standard for MIC testing. Recently, two commercial kits (Compact Antimicrobial Susceptibility Panel, ComASP, Liofilchem, Italy and UMIC, Bruker, Germany) to determine cefiderocol MICs through BMD have been released on the market. Data on the diagnostic performances of these new assays with *A. baumannii* are scarce.

While the BMD method is labor-intensive and time-consuming, disk diffusion (DD) and/or E-test are quick and easy to perform and are commonly used in most laboratories. It is important to note that, while for MIC determination with the standard BMD method both EUCAST and CLSI recommend the use of ID-CAMHB, the breakpoints released for DD are based on the use of CAMH-agar-plates. Perhaps because of this inconsistency, several works have reported poor performances of DD, especially when MIC values were distant from the breakpoints [6,7].

Resistance against cefiderocol in *A. baumannii* can arise from a plethora of different mechanisms: plasmid-borne β-lactamases such as PER-, SHV-type ESBLs, and NDM-type carbapenemases [8,9], mutations affecting the expression and function of the intrinsic AmpC-type-β-lactamase ADC, the siderophore importers, and to a lesser extent, porins [10,11] and mutations that alter the PBP3 target gene of cefiderocol [12]. Moreover, recent reports indicate that upon exposure to high cefiderocol concentrations, *A. baumannii* may implement adaptation mechanisms that can give rise to hetero-resistant, unstable subpopulations, which may be difficult to detect, also with the standard BMD method [13,14]. Also, the high prevalence of heteroresistance to cefiderocol has been proposed as an explanation for the suboptimal efficacy of cefiderocol against CRAB [15]. As a result, accurate susceptibility testing is key for the success of cefiderocol treatment, especially considering the higher prevalence of resistance among CRAB compared to carbapenem-susceptible *A. baumannii* isolates [16].

Several studies have shown that avibactam, as well as other β-lactamase-inhibitors, can restore susceptibility in cefiderocol-resistant *A. baumannii* isolates producing OXA-type and/or serine (i.e., PER-type) β-lactamases, as well as in isolates exhibiting cefiderocol heteroresistance [8,9,14,17,18]. However, an adequate test to detect synergism in the microbiology diagnostic lab is currently unavailable.

In this study, we aimed to evaluate the diagnostic performances of two commercial BMD assays, ComASP and UMIC, and the DD and E-tests on standard MH-agar and ID-MH-agar. We also aimed to validate a DD- and E-test-based method for detection of synergistic activity of cefiderocol with avibactam. To this end, we used a collection of 100 CRAB isolates producing a wide range of resistance mechanisms and showing a broad spectrum of cefiderocol susceptibility profiles.

## 2. Materials and Methods

### 2.1. Strain Collection

A total of 100 *A. baumannii* isolates were used (Appendix A). Fourteen were obtained from the Institute Pasteur’s strain collection (https://www.pasteur.fr/en/public-health/biobanks-and-collections/collection-institut-pasteur-cip, accessed on 14 March 2023), while the remaining eighty-six clinical isolates were collected from individual patients between January 2014 and December 2022 in the routine diagnostic laboratory of the Institute of Medical Microbiology at the University of Zurich.

### 2.2. MALDI-TOF MS Identification

Bacterial isolates were prepared for matrix-assisted laser desorption ionization–time of flight mass spectrometry (MALDI-TOF MS) by the direct transfer-formic acid method [19]. Species identification was performed using the Bruker Biotyper MALDI-TOF MS System (Bruker corporation, Billerica, MA, USA).

### 2.3. Β. Eta-Lactam AST

Cefiderocol susceptibility was determined by BMD, DD, and the E-test gradient strip method (see the next paragraphs). DD was performed to determine the susceptibility toward classic β-lactams (piperacillin-tazobactam, ceftazidime, cefepime, meropenem, and imipenem), aminoglycosides (amikacin, gentamicin, and tobramycin) and quinolones (ciprofloxacin and levofloxacin, see Appendix A). Antibiotic disks were from Oxoid Limited (Basingstoke, United Kingdom). Sirweb/Sirscan system (i2a) measured the inhibition zone diameters, which were visually controlled [20]. The gradient strip test (E-test) was used to determine MICs of ceftazidime-avibactam, ceftolozane-tazobactam, ampicillin-sulbactam, tigecycline, and eravacycline (Appendix A). All antibiotic gradient strips were from Liofilchem. Colistin MICs were determined by BMD using the UMIC Colistin kit (Biocentric, Bandol, France). MIC values were rounded and adjusted to a binary log scale (i.e., 0.002, (…), 128, 256). *Pseudomonas aeruginosa* ATCC 27853 and *A. baumannii* NCTC13304 were used as internal quality control strains to assess the accuracy and reproducibility of different AST methods (Appendix A).

### 2.4. BMD for Cefiderocol AST

Cefiderocol BMD was performed in different setups: first, according to EUCAST guidelines using iron-depleted cation-adjusted Mueller-Hinton (ID-CAMH) medium; second, using two commercial kits, ComASP^®^ Cefiderocol 0.008–128 µg/mL (Liofilchem, Roseti degli Abruzzi, Italy) and UMIC^®^ Cefiderocol (Biocentric, Bandol, France) [21]. Results were interpreted based on the CLSI clinical breakpoints (CBPs) and the EUCAST PK-PD breakpoint [3,4]. ID-CAMHB was prepared according to CLSI approved methodology: [22] 50 g of Chelex^®^ 100 resin (Bio-Rad Laboratories, Hercules, CA, USA) were added to 1 L of autoclaved CAMHB (Merck KGaA, Darmstadt, Germany). The suspension was stirred for 2 h at room temperature (23 °C) to remove cations in the medium. The iron-depleted broth was passed through a 0.2 μm filter to remove the resin. The pH of the broth was adjusted to 7.3 using 0.1 M hydrochloric acid. The ID-CAMHB was supplemented with 22.5 μg/mL CaCl_2_ (range, 20–25 μg/mL), 11.25 μg/mL MgCl_2_ (range, 10–12.5 μg/mL), and 10 μM ZnSO_4_ (0.56 μg/mL; range 0.5–1.0 μg/mL). The solution was finally passed through a 0.2 μm filter for sterilization. For synergy testing, all cefiderocol concentrations were supplemented with a fixed concentration of 4 µg/mL avibactam. Cefiderocol and avibactam were from MedChemExpress (Monmouth Junction, NJ, USA). Synergistic activity was defined as a decrease in cefiderocol MICs induced by avibactam of three or more-fold-dilutions.

### 2.5. DD for Cefiderocol AST

The Kirby-Bauer DD method was performed on CAMH-agar plates (BioMérieux, Marcy L’Etoile, France) according to EUCAST guidelines [23]. In addition, cefiderocol DD was performed on CAMH-agar plates from a different manufacturer (Liofilchem, Roseti degli Abruzzi, Italy) and on in-house produced ID-CAMH-agar plates. For the preparation of ID-CAMH-agar plates, 20 g of agar were added to 1 L CAMHB (see the previous section). After sterilization, 25 mL media were dispensed in empty Petri dishes. Cefiderocol disks (30 µg) were from Liofilchem. For synergy testing of ceftazidime-avibactam with cefiderocol, disks containing ceftazidime-avibactam 10 + 4 μg (CZA14), cefiderocol 30 μg, and ceftazidime-avibactam 40 + 10 μg (CZA50) were placed on in-house produced ID-CAMH-agar plates at distance of 2 cm. After 18 h incubation at 35 °C, the zone inhibition between CZA14, CZA50, and cefiderocol was visually inspected and documented.

### 2.6. MIC Test Strip for Cefiderocol AST

The MIC test strip method was performed on regular CAMH-agar plates from BioMérieux (Marcy L’Etoile, France). Cefiderocol MIC test strip was also performed on CAMH-agar plates from Liofilchem and on in-house produced ID-CAMH-agar plates. For synergy testing of ceftazidime-avibactam with cefiderocol, the ceftazidime-avibactam strip was first applied on an in-house produced ID-CAMH-agar plate previously inoculated with *A. baumannii* (McFarland 0.5) using a swab. After 1 h incubation at room temperature, the ceftazidime-avibactam strip was carefully removed and the cefiderocol strip was placed on the same location. Cefiderocol MICs were read after 18 h incubation at 35 °C.

### 2.7. Data Analysis

Essential agreement (EA) was defined as MIC ± one twofold dilution of the reference MIC (determined with the reference BMD method). Categorical agreement (CA) and clinical errors (major error, ME; very major error, vME) were determined according to the ‘CLSI Methods Development and Standardization Working Group Best Practices for Evaluation of Antimicrobial Susceptibility Tests (2018)’ on the basis of the CLSI and EUCAST breakpoints [3,4]. Expected congruent performances were: EA/CA ≥ 90%, ME ≤ 5%, vME ≤ 1.5% [24].

### 2.8. Whole Genome Sequencing and Typing

Whole genome sequencing determined the presence of beta-lactamase genes. Bacterial genomic DNA was extracted using the DNeasy^®^ Ultraclean^®^ Microbial kit (Qiagen, Hilden, Germany). Library preparation was performed with the QIAGEN QIASeq FX kit (Qiagen, Hilden, Germany). Library quality and fragment size distribution were analyzed on an automated CE system (Advanced Analytical Technologies Inc., Heidelberg, Germany). Paired-end sequencing (2 × 150 bp) of DNA libraries was performed using an Illumina MiSeq platform (Illumina^®^, San Diego, CA, USA). Trimmomatic (version 0.39) was used to filter and trim raw sequencing data [25]. Reads were assembled using Unicycler v0.4.8 [26]. All genome assemblies were typed in Ridom Seqsphere+ v8.5.1 by multilocus sequence typing (MLST) according to the Pasteur (sequence type, ST) scheme and by core genome multi-locus sequence typing (MLST) [27].

### 2.9. Detection of β-Lactam Resistance Genes

Plasmidic β-lactamase-genes were identified within assemblies querying the NCBI database [28] using abricate (https://github.com/tseemann/abricate, accessed on 14 March 2023), or through Resfinder 4.1 [29]. All read data were submitted to the ENA (https://www.ebi.ac.uk/ena/browser, accessed on 14 March 2023) under project number PRJEB62871.

## 3. Results

### 3.1. Antimicrobial Susceptibility and Genomic Analysis

The range of cefiderocol MICs determined with the reference BMD method was 0.125 to >64 μg/mL, the MIC_50_ 1 μg/mL, and MIC_90_ 4 μg/mL (Appendix A). Based on CLSI breakpoints (S ≤ 4 μg/mL, R > 8 μg/mL), 83 isolates were classified as susceptible, 5 intermediate and 12 resistant (Figure 1). Based on the EUCAST PK-PD breakpoint (S ≤ 2 μg/mL, R > 2 μg/mL), 76 isolates were susceptible and 24 resistant.

Genomic analysis revealed the presence of plasmid-borne carbapenemases in 94/100 isolates (Appendix A). The great majority were class D carbapenemases of type OXA-23 (63%), followed by OXA-72 (14%), OXA-58 (3%), and OXA-40 (1%). In four isolates (4%), a class B carbapenemase of type NDM was found (three NDM-1 and one NDM-2). In eight isolates (8%), a combination of different oxacillinase families (one with OXA-23/OXA-58 and two with OXA-23/OXA-72) or an oxacillinase in combination with class D carbapenemase (two NDM-1/OXA-23 and three NDM-1/OXA-72) were detected. In one isolate (1%), the class A carbapenemase GES-14 was detected. Plasmid-borne ESBL GES-11 was found in two carbapenemase-negative isolates (2%), while in none of the remaining four isolates (4%) plasmid-borne resistance markers associated with carbapenem-resistance were found.

The *A. baumannii* isolates belonged to 34 different sequence types (STs), one of which was novel and four were not determined (Appendix A). ST2 was the most prevalent ST (49/100, 49%). The phylogeny falls into two major clusters, one dominated by ST2. This phylogenetic organization has previously been described [30]. The dominance of ST2 and, in particular, one sublineage of ST2, reflects what is received by our laboratory in Zürich, Switzerland. No association between cefiderocol resistance and phylogenetic clusters was detected, as resistant isolates are often adjacent to sensitive isolates in the phylogeny, suggesting independent emergence of cefiderocol resistance through gene acquisition and de novo mutations (Appendix A).

The *P. aeruginosa* ATCC27853 quality control (QC) strain was tested throughout the experiments (eight times) and exhibited MIC values within the EUCAST range (0.06–0.5 μg/mL, MIC mean 0.22 ± 0.1 μg/mL), while the *A. baumannii* NCTC13304 QC strain, for which there are neither EUCAST nor CLSI DD QC MIC range values, exhibited a MIC mean of 0.62 ± 0.2 μg/mL (Appendix A).

Based on the EUCAST and CLSI CBPs for *A. baumannii*, and when not available for *P. aeruginosa*, nearly all the isolates were resistant towards piperacillin-tazobactam, cephalosporins (ceftazidime and cefepime), carbapenems (imipenem and meropenem), and quinolones (ciprofloxacin and levofloxacin, see Appendix A and Appendix A). Also, the great majority displayed resistance against all classic aminoglycosides (amikacin, gentamicin, and tobramycin). As expected, all isolates showed high MICs of ceftazidime-avibactam, ceftolozane-tazobactam, and ampicillin-sulbactam (for all MIC90 > 256 μg/mL), irrespective of their β-lactamase content (Appendix A). For tigecycline and eravacycline, there are no EUCAST nor CLSI CBPs on *A. baumannii* or *P. aeruginosa* (EUCAST has so far published a CBP for *E. coli*, which is 0.5 μg/mL for both tigecycline and eravacycline). The MIC90 were 4 and 1 μg/mL, respectively. Only four isolates exhibited high colistin MICs (64–128 μg/mL), while the remaining strains showed MICs in the susceptible range (MIC90 = 1 μg/mL).

### 3.2. Performances of Disk Diffusion to Assess Cefiderocol Susceptibility

Disk diffusion was performed on two commercially available CAMH-agar plates (BioMérieux and Liofilchem) and on an in-house produced ID-CAMH-agar plate. Results were compared with MICs determined with the reference BMD method (Figure 2). Based on CLSI guidelines, whereby cefiderocol susceptibility of *A. baumannii* should be reported only when the inhibition zone is bigger than 14 mm, with both commercial CAMH-plates more than 90% (BioMérieux 90/100, Liofilchem 93/100) of the isolates were classified as susceptible (inhibition zone > 14 mm). While using the in-house produced ID-CAMH-agar plate, only 81% (81/100, Table 1) were susceptible. However, considering only the interpretable results, the CA with the reference BMD method was higher with the ID-CAMH-agar plate (77/81, 95.1%) than with the commercial CAMH-plates (BioMérieux 82/90, 91.1%; Liofilchem 83/93, 89.2%). Furthermore, the ID-CAMH-agar plate caused significantly less categorization errors (3/81 mE, 1/81 vME) as compared to CAMH-agar plates (BioMérieux 4/90 mE, 4/90 vME; Liofilchem 4/93 mE, 6/93 vME). Based on the EUCAST PK-PD breakpoint (S ≥ 17 mm, R < 17 mm), the CA was 87%, 84%, and 86% with the BioMérieux-, Liofilchem- and ID-CAMH-agar plates, respectively. Again, the ID-CAMH plates generated less vME (7/100) compared to the plain CAMH-agar plates (BioMérieux 11/100; Liofilchem 16/100).

The *P. aeruginosa* ATCC27853 QC strain exhibited DD values within the EUCAST range (23–29 mm) when using the CAMH-agar plates (both showing a mean growth inhibition zone of 28 ± 1 mm). While the mean inhibition zone on the homemade ID-CAMH-agar plates was slightly bigger (29.8 ± 1.5 mm) and was in three cases above the higher range value (Appendix A). The growth inhibition zones of the *A. baumannii* NCTC13304 QC strain were stable throughout the experiments. The inhibition zone varied in size based on the media used: with the Liofilchem-CAMH-agar plate of 25 ± 0.7 mm, with the BioMérieux-CAMH-agar plate of 23.6 ± 0.9 mm, and with the ID-CAMH-agar plate of 21 mm.

### 3.3. Performances of E-Test to Assess Cefiderocol Susceptibility

The E-test method was performed on two commercial CAMH-agar plates (BioMérieux and Liofilchem) and on an in-house produced ID-CAMH-agar plate. Results were compared with MICs determined with the reference BMD method (see Table 1, Figure 3). The ID-CAMH-agar plates showed superior EA with the reference BMD method (75/100, 75%), as compared to CAMH-agar plates from BioMérieux (57/100, 57%) and Liofilchem (44/100, 44%). Based on CLSI guidelines (S ≤ 4 µg/mL, I = 8 µg/mL, R ≥ 16 µg/mL), the CA with the reference BMD method for ID-CAMH-agar plates (87/100, 87%) was comparable with that for the commercial CAMH-agar plates (BioMérieux, 85/100, 85%; Liofilchem CA = 88/100, 88%). However, ID-CAMH-agar plates caused significantly fewer vMEs than the others (ID-CAMH-agar 2/100, 2% vMEs; BioMérieux CAMH-agar 10/100, 10% vMEs; Liofilchem CAMH-agar 7/100, 6.9% vMEs). Consistent with these findings, the mean log_2_ difference of the MICs with the standard BMD of the intermediate and resistant populations was 0.5 with ID-CAMH-agar plates, while 3.58 and 2.83 with the BioMérieux and Liofilchem CAMH-agar plates, respectively (Appendix A). According to EUCAST PK-PD breakpoint (S ≤ 2 µg/mL, R > 2 µg/mL), the CA with the reference BMD method was comparable with all three media, whereas ID-CAMH-agar plates once again generated the lowest number of vMEs (5/100, 5% as compared to CAMH-agar plates (BioMérieux and Liofilchem), which produced 12/100 (12%) and 13/100 (13%) vMEs, respectively.

The *P. aeruginosa* ATCC27853 QC strain exhibited comparable MIC values when using the three different CAMH-agar plates, and all the growth inhibition values were within the QC range (Appendix A). The MICs of the *A. baumannii* NCTC13304 QC strain were stable throughout the experiments and were identical when using the commercial CAMH-agar plates (mean 0.12 µg/mL), while being 1–2 log_2_ higher when using the ID-CAMH-agar plate (mean 0.41 µg/mL).

### 3.4. Performances of ComASP to Assess Cefiderocol Susceptibility

ComASP showed 76% EA with the standard BMD method (Table 1, Figure 3 and Appendix A). According to CLSI and EUCAST guidelines, ComASP exhibited 86% and 88% CA with the reference method and produced vME with 6/100 and 7/100 isolates, respectively.

MIC values of the *P. aeruginosa* ATCC27853 QC strain were all within the EUCAST QC range and the MIC mean (0.33 ± 0.1 µg/mL) was comparable to that of the standard BMD method (0.22 ± 0.1 µg/mL). Likewise, MICs of the *A. baumannii* NCTC13304 QC strain as determined with ComASP (MIC mean 0.53 ± 0.3 µg/mL) were comparable to those assessed with the BMD reference method (MIC mean 0.62 ± 0.2 µg/mL).

### 3.5. Performances of UMIC to Assess Cefiderocol Susceptibility

As for ComASP, the UMIC test showed 76% EA with the standard BMD method (Table 1, Figure 3 and Appendix A). According to CLSI and EUCAST guidelines, UMIC exhibited 86% and 89% CA with the reference method, and produced vME with 3/100 and 9/100 isolates, respectively.

MIC values of the *P. aeruginosa* ATCC27853 QC strain were all within the EUCAST QC range and the MIC mean (0.23 ± 0.2 µg/mL) was comparable to that of the standard BMD method (0.22 ± 0.1 µg/mL). Instead, the MICs of the *A. baumannii* NCTC13304 QC strain as determined with UMIC (MIC mean 0.31 ± 0.1 µg/mL) were on average one log_2_ lower than those assessed with the BMD reference method (MIC mean 0.62 ± 0.2 µg/mL).

### 3.6. Overall Performances of the Various Methods to Assess Cefiderocol Susceptibility

Overall, based on the CLSI CBPs and considering the CA, DD performed better than the MIC-based methods (DD CA = 89.2–95.1%, E-test, ComASP and UMIC CA = 85–87%) and DD with ID-MH-plates exhibited the highest congruence with the BMD method (CA = 95.1%, see Table 1). Importantly, using ID-MH-plates led to significantly less vME both by DD (1.2%) and by E-test (2%) as compared with the other commercially available MH-plates (≥4.4%), mostly due to detection of cefiderocol highly resistant isolates. UMIC also produced few vME (3%) as compared to the other MIC-based methods (≥6%). Among the MIC-based methods, the E-test with ID-MH-plates, ComASP and UMIC exhibited the highest EA with the standard BMD method (75–76%). However, ComASP produced significantly more vMEs (6% vs. 2–3%) as a result of the failure to detect highly resistant isolates.

### 3.7. Synergy between Cefiderocol and Avibactam

We found that addition of avibactam decreased the cefiderocol MICs by three- or more-fold dilutions (synergistic activity) and restored in vitro susceptibility in 3/5 intermediate and all 9 resistant *A. baumannii* strains non-producing MBL-carbapenemases (i.e., of type NDM) and exhibiting cefiderocol MICs ≥ 8 mg/L (Table 2). Interestingly, in one cefiderocol intermediate (isolate 30, OXA-58-producer) and one resistant (isolate 92, OXA-23/-72-producer) *A. baumannii* strain, the addition of avibactam did not affect cefiderocol MICs as determined by standard BMD. Synergy tests using the MIC gradient strip method exhibited concordant data with the BMD method in all but one *A. baumannii* strains (see an explanatory example on Figure 4). The one discordant *A. baumannii* isolate (isolate 73, OXA-23-producer) tested cefiderocol resistant with the BMD method (MIC = 16 µg/mL) but resulted susceptible with the MIC gradient strip method (MIC = 0.75 µg/mL). Also, cefiderocol susceptibility was not affected by avibactam. The two *A. baumannii* strains for which with the standard BMD method avibactam did not show synergistic activity with cefiderocol (isolates 30 and 92), neither a synergistic effect nor restoration of cefiderocol susceptibility was observed with the MIC gradient strip test. Finally, growth inhibitory effects (halos) between cefiderocol and avibactam disks (either ceftazidime/avibactam 10/4 µg and/or ceftazidime/avibactam 40/10 µg, see an explanatory example on Figure 4) were detected by DD with 8/10 *A. baumannii* isolates showing synergistic activity with the BMD method, while it was not detected in the remaining two *A. baumannii* strains (isolates 57 and 90). 

## 4. Discussion

Using a large collection of CRAB with a wide range of cefiderocol susceptibilities, we showed that DD and E-test performed on CAMH-agar plates exhibit a poor correlation with the standard BMD. Importantly, both methods tend to underestimate MICs, especially with highly resistant strains. Likewise, we found that the recently commercialized ComASP, and to a lesser extent, UMIC, also failed to detect high-level resistance, mostly because of underestimation of high MICs, even though UMIC exhibited a higher congruence with the standard BMD method. The congruence with the standard BMD values significantly increased when both DD and E-test were performed with the same medium, namely ID-CAMHB. Like for the exemplary isolate depicted on Appendix A, tiny yet visible colonies within the growth inhibition zones of resistant isolates, which may result from the emergence of hetero-resistant subpopulations, appeared more consistently on ID-CAMH agar plates. This improved the correlation with the MIC values obtained with the standard BMD method. Consistent with these findings, the ID-CAMH-agar plates exhibited a significantly lower mean log_2_ difference of MICs with the standard BMD of the intermediate and resistant populations as compared to the other methods (Appendix A). To our knowledge, this is the first study evaluating the performance of DD and E-test for cefiderocol and *A. baumannii* using the ID-CAMH-agar plates.

To improve the reliability of the cefiderocol ASTs, two tests instead of one may be performed and interpreted. For example, considering the DD and E-test values obtained on ID-CAMH agar plates (interpreted according to the CLSI CBPs and based on the rule that by discrepant categorization between the methods resistant overtake susceptible results), CA with the BMD method was observed in in 88/100 of the cases, mE in 9/100, ME in 1/100 and vME in 2/100 (Appendix A). Applying the same rules and considering the DD values obtained on ID-CAMH agar plates and the UMIC MIC values, CA with the BMD method was observed in in 89/100 of the cases, mE in 8/100, and vME in 3/100 cases. Finally, considering the DD values obtained on ID-CAMH agar plates and the ComASP MIC values, CA with the BMD method was observed in 88/100 of the cases, and mE and vME both in 6/100 cases.

Repeated testing showed that the medium had no impact on the cefiderocol E-test values and had only a small impact on the DD values of the *P. aeruginosa* ATCC27853 QC strain. Conversely, the *A. baumannii* NCTC13304 QC strain exhibited on average significantly higher MICs and smaller inhibition zones when using the ID-CAMH-agar plates, suggesting a bigger impact of the medium on the cefiderocol AST of susceptible *A. baumannii* strains. Moreover, the ID-CAMH-agar plates improves the detection of putative resistant subpopulations. A set of two *A. baumannii* QC strains, one susceptible and one resistant to cefiderocol, may also be considered for internal QC to ensure the quality of the CAMH-agar-plates.

Previous studies have shown that PER-like β-lactamases, and to a lesser degree, NDM β-lactamases, are associated with elevated MICs in *A. baumannii*, although production of these enzymes alone does not lead to MICs above the EUCAST PK-PD breakpoint (≤2 µg/mL) [9]. Likewise, in our study all PER-producing strains exhibited high MIC values (≥32 µg/mL), while NDM-producing isolates showed either reduced susceptibility or resistance with MIC values closely above to the CBPs. Cefiderocol reduced susceptibility and/or resistance was not associated with specific markers. Thus, cefiderocol susceptibility cannot be inferred by the presence specific acquired resistance mechanisms and should always be determined in vitro. The only exception to this rule is for PER-type β-lactamases, whose prompt detection may help guide decision making in therapy against MDR *A. baumannii* infections. In this regard, we showed that the addition of avibactam restored the susceptibility in all but one cefiderocol resistant *A. baumannii* isolate producing OXA-type and/or PER-type β-lactamases, as also reported in previous studies [8,9,10,14,17,18]. We also found that, using ID-CAMH-agar plates, the synergistic activity of avibactam with cefiderocol can be tested quantitatively and qualitatively by E-test and DD, respectively.

In conclusion, we showed that DD and E-test on ID-CAMH-agar plates can produce more consistent results with the standard BMD method than on CAMH-agar plates, which are currently recommended by EUCAST and CLSI. Synergy between cefiderocol and avibactam can also be detected both by E-test and DD on ID-CAMH-agar plates. Based on the findings of this study, ID-CAMH-agar plates may be considered for in vitro susceptibility testing of cefiderocol and *A. baumannii*.

## Figures and Tables

**Figure 1 antibiotics-12-01212-f001:**
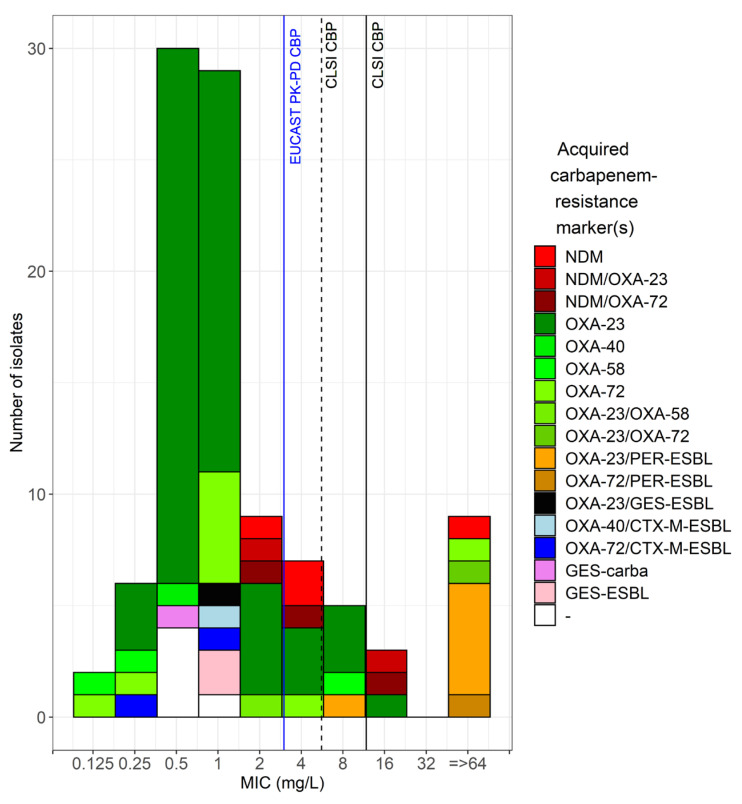
Distribution of cefiderocol MICs determined with the standard BMD method according to the acquired carbapenem-resistance marker(s). The vertical lines denote the CLSI CBPs for *A. baumannii* (dashed and continuous black) and the EUCAST PK-PD breakpoint (blue). MIC reading was performed according to the EUCAST guidance document on broth microdilution testing of cefiderocol.

**Figure 2 antibiotics-12-01212-f002:**
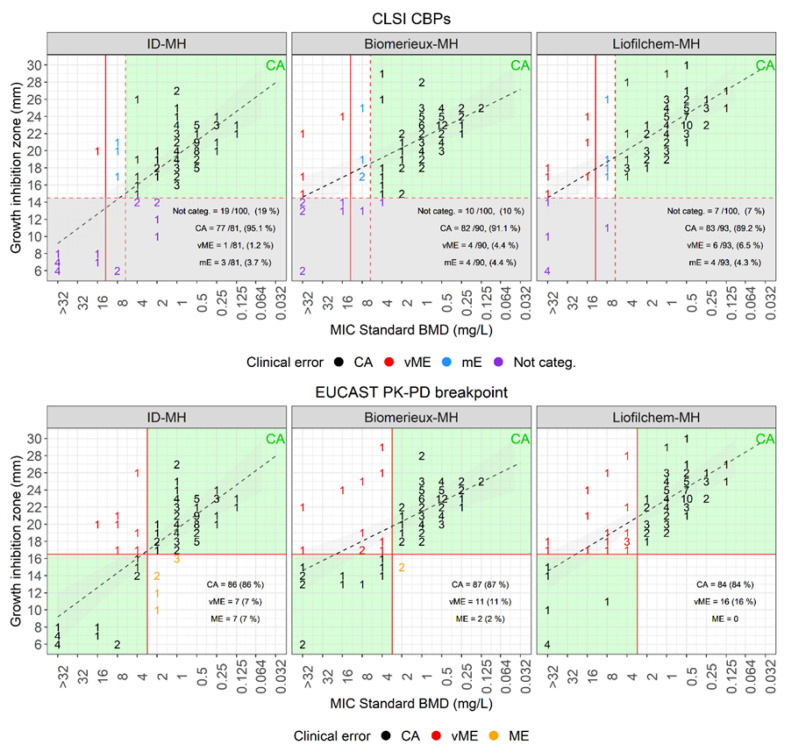
DD versus standard BMD. Cefiderocol disk diffusion growth inhibition zones on iron-depleted MH-agar (ID-MH), BioMérieux MH-agar (BioMerieux-MH), and Liofilchem MH-agar (Liofilchem-MH) versus MICs determined with the standard BMD method. BMD MICs are on the X-axis and zone diameters on the Y-axis. Isolates were categorized according to the BMD MICs and CLSI CBPs (**top figures**) or EUCAST PK-PD breakpoint (**bottom figures**). MICs/zone diameters were classified as categorical agreement in black, very major error in red, major error in orange, minor error in blue, and not categorizable in violet. The red dashed and continuous lines denote the CLSI CBPs (**top**) and the EUCAST PK-PD breakpoint (**bottom**). The black dashed lines denote the regression lines. The green areas denote zones of congruence between the two methods, while the gray areas denote the area where the zone inhibition diameter cannot be categorized (when <15 mm, according to the CLSI guidelines.).

**Figure 3 antibiotics-12-01212-f003:**
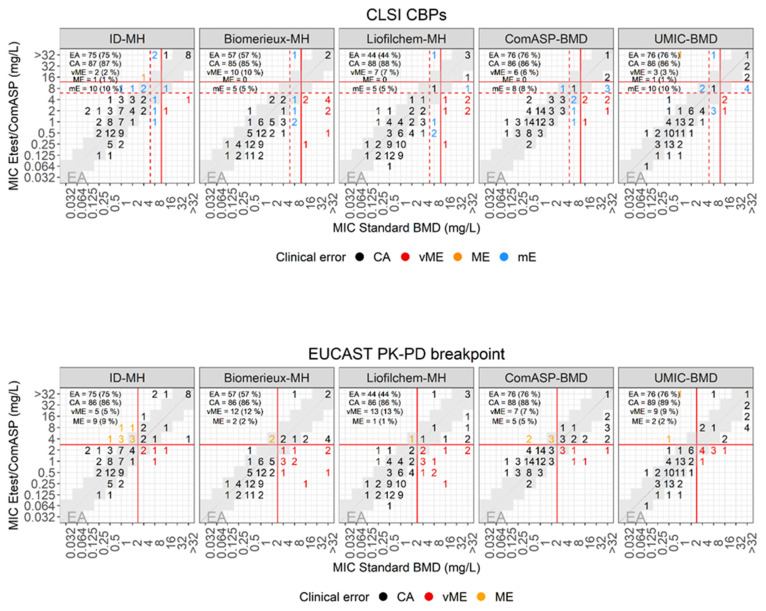
E-test versus standard BMD. MICs as determined by E-test on iron-depleted MH-agar (ID-MH), BioMérieux MH-agar (BioMerieux-MH), Liofilchem MH-agar (Liofilchem-MH), and with the commercial BMD assays ComASP (Liofilchem) and UMIC (Bruker) versus MICs determined with the standard BMD method. Standard BDM MICs are on the x axis and E-test/ComASP/UMIC MICs on the y axis. Isolates were categorized based on the BMD MICs and the CLSI CBPs (**top figures**) or the EUCAST PK-PD breakpoint (**bottom figures**). Standard MICs/E-test or ComASP/UMIC MICs were classified as categorical agreement in black, very major error in red, major error in orange, and minor error in blue. The red dashed and continuous lines denote the CLSI CBPs (**top**) and the EUCAST PK-PD breakpoint (**bottom**). The gray highlighted areas denote essential agreement (MIC ± one two-fold dilution of the reference MIC).

**Figure 4 antibiotics-12-01212-f004:**
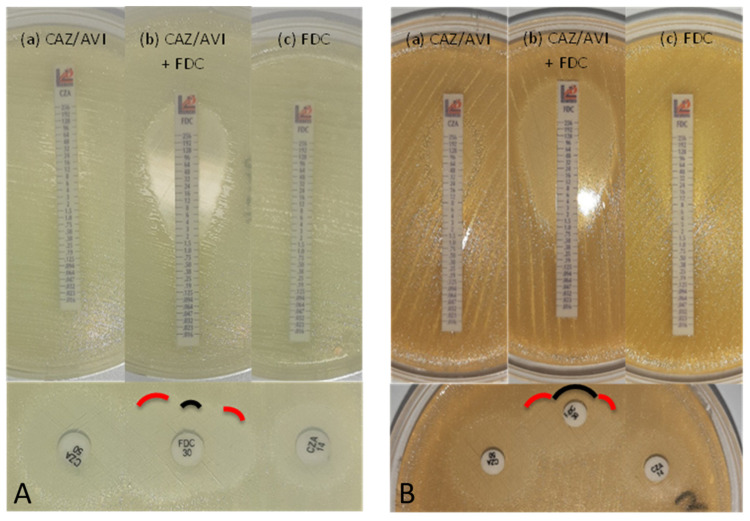
Examples of synergistic combination of ceftazidime/avibactam (CZA) and cefiderocol (FDC) for an OXA-23-producing A. baumannii isolate. (**A**) isolate 20, (**B**) isolate 92. On top are displayed the E-test gradient strip tests of CZA alone (a), CZA with FDC (b) and FDC alone (c). On the bottom is displayed the double disk synergy test with disks containing ceftazidime/avibactam 40 + 10 μg (CZA50), cefiderocol 30 μg (FDC30) and ceftazidime/avibactam 10 + 4 μg (CZA14).

**Table 1 antibiotics-12-01212-t001:** Diagnostic performances of Cefiderocol DD, E-test, and ComASP.

Disk Diffusion Versus Standard BMD									
Plate	Breakpoint (mm)				Categorized, ≥ 15 mm (%)	Not Categorized (%)	CA (%)	mE (%)	vME (%)
	S≥		R<	Source					
MH-BioMérieux	15			CLSI	90 (90)	10 (10)	82/90 (91.1)	4 (4.4)	4 (4.4)
MH-Liofilchem					93 (93)	7 (67)	83/93 (89.2)	4 (4.3)	6 (6.5)
ID-MH-homemade					81 (81)	19 (19)	77/81 (95.1)	3 (3.7)	1 (1.2)
Plate	Breakpoint (mm)						CA (%)	ME (%)	vME (%)
	S≥		R<	Source					
MH-BioMérieux	17		17	EUCAST PK-PD			87 (87)	2 (2)	11 (11)
MH-Liofilchem							84 (84)	0 (0)	16 (16)
ID-MH-homemade							86 (86)	7 (7)	7 (7)
E-test versus standard BMD									
Plate	Breakpoint (mm)				EA (%)	CA (%)	mE (%)	ME (%)	vME (%)
	S≤	I =	R>	Source					
MH-BioMérieux	4	8	8	CLSI	57 (57)	85 (85)	5 (5)		10 (10)
MH-Liofilchem					44 (44)	88 (88)	5 (5)		7 (7)
ID-MH-homemade					75 (75)	87 (87)	10 (10)	1 (1)	2 (2)
ComASP					76 (76)	86 (86)	8 (8)		6 (6)
UMIC					76 (76)	86 (86)	10 (10)	1 (1)	3 (3)
Plate	Breakpoint (mm)				EA (%)	CA (%)	mE (%)	ME (%)	vME (%)
	S≤	I =	R>	Source					
MH-BioMérieux	2		2	EUCAST PK-PD	57 (57)	86 (86)		2 (2)	12 (12)
MH-Liofilchem					44 (44)	86 (86)		1 (1)	13 (13)
ID-MH-homemade					75 (75)	86 (86)		9 (9)	5 (5)
ComASP					76 (76)	88 (88)		5 (5)	7 (7)
UMIC					76 (76)	89 (89)		2 (2)	9 (9)

**Table 2 antibiotics-12-01212-t002:** Synergistic activity of cefiderocol and avibactam in cefiderocol-resistant *A. baumannii* isolates.

Method		Standard BMD, MIC (μg/mL)			E-Test on ID-MH-Agar, MIC (μg/mL)				Double Disk Diffusion on ID-MH-Agar, Inhibition Zone (mm)		
Isolate n.	Major Plasmidic β-Lactamase(s)	CFD	CFD + AVI ^1^	Fold Difference	CZA	CFD	CFD + CZA	Fold Difference	CFD	CZA14	CZA50
9	OXA-72	>32	2	>4	32	>256	0.5	>9	6	6	14
22	OXA-23/PER-1	8	0.5	4	32	1	0.125	3	19	10	16
25	OXA-23/PER-1	>32	0.5	>6	16	>256	0.125	>11	12	12	17
30	OXA-58	8	4	1	>256	6	2	1.5	14	6	11
45	OXA-72/PER-1	>32	1	>5	32	>256	1	>8	6	11	19
56	OXA-23/PER-7	>32	1	>5	16	>256	0.19	>10	6	10	15
57	OXA-23/PER-7	>32	1	>5	96	>256	2	>7	6	8	14
69	OXA-23/PER-7	>32	1	>5	16	12	1	3.5	10	13	18
73	OXA-23	16	2	3	48	0.75	0.38	1	23	8	15
78	OXA-23/PER-7	>32	1	>5	24	16	0.125	7	8	12	17
85	OXA-23	8	1	3	>256	3	0.38	3	18	6	14
90	OXA-23	8	0.0625	7	192	>256	1.5	>7	6	6	12
92	OXA-23/OXA-72	≥32	≥32	0	>256	>256	32	>3	6	6	8
95	OXA-23	8	0.125	6	64	3	0.5	3	18	8	14

CFD, cefiderocol; AVI, avibactam; CZA, ceftazidime/avibactam; CZA14, ceftazidime 10 µg/avibactam 4 µg; CZA50, ceftazidime 40 µg/avibactam 10 µg. The green background indicates the detection of haloes (synergistic activity) between the CZA and CFD disks.

## Data Availability

All WGS read data are available in the ENA website (https://www.ebi.ac.uk/ena/browser, accessed on 14 March 2023) under project number PRJEB62871.

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
