# Peer review of "Comparison of Disk Diffusion, E-Test, and Broth Microdilution Methods for Testing In Vitro Activity of Cefiderocol in Acinetobacter baumannii"

_antibiotics, 2023, doi:10.3390/antibiotics12071212_

Round 1
Reviewer 1 Report
The authors would like to compare the various methods for determining the susceptibility of carbapenem-resistant A. baumannii against cefiderocol. Although the methods are performed based on commercial protocol and CLSI/EUCAST recommendations, but the results are poorly presented. The use of whole genome sequencing to further support the study is also lack of analysis. All the results need to be analyzed as a whole and written in a simple and clear manner.
Some other comments are highlighted below:
1. Please provide citation for line 32-46 in order to support the claims.
2. Line 72, the use of genetical diverse strains may not be appropriate since majority of the strains belong to ST2. Unless, the authors perform genomic comparison to provide the evidence.
3. Line 157, the project number PRJEB62871 was not found in ENA. Please make sure all the genomes are publicly available such as in NCBI.
4. Line 160-170, please describe in terms of percentage e.g. how many % contains OXA-23.
5. There is a lack of explanation on the whole genome sequencing data.
6. For Figure S4, please explain why the phylogenetic tree has two major branches.
7. Line 175-177, please clarify the statement by providing data.
8. Line 2-37 in page 10 is the repeat of line 184-219. Please check the error.
9. Line 41, the authors will need to perform additional tests to confirm the existence of subpopulations.
No major comments.
Author Response
The authors would like to compare the various methods for determining the susceptibility of carbapenem-resistant A. baumannii against cefiderocol. Although the methods are performed based on commercial protocol and CLSI/EUCAST recommendations, but the results are poorly presented. The use of whole genome sequencing to further support the study is also lack of analysis. All the results need to be analyzed as a whole and written in a simple and clear manner.
WGS data were analyzed for bla genes and core genome MLST to assess the diversity, and the results were described in the ‘Results’ section (lines 212-226). Also, a paragraph was added in the ‘Results’ section where the phenotypic results have been analyzed in a simple and clear manner (lines 179-193).
Some other comments are highlighted below:
- Please provide citation for line 32-46 in order to support the claims.
Done.
- Line 72, the use of genetical diverse strains may not be appropriate since majority of the strains belong to ST2. Unless the authors perform genomic comparison to provide the evidence.
As can be seen in Figure S4, there is substantial genomic diversity within these strains. However, ‘genomic diverse’ and ‘genetical diverse’ have been removed from the text (lines 83 and 91).
- Line 157, the project number PRJEB62871 was not found in ENA. Please make sure all the genomes are publicly available such as in NCBI.
The project had not been released yet. Now it is publicly available in ENA under the number PRJEB62871.
- Line 160-170, please describe in terms of percentage e.g. how many % contains OXA-23.
Done.
- There is a lack of explanation on the whole genome sequencing data.
The strains were primarily sequenced to confirm the presence of relevant bla genes, as given in the results and Figure 1. Additionally, we created a core gene phylogeny and extracted STs from the data, giving results compatible with those previously published. We have clarified this in the text (lines 212-226). This is given in the text and Figure S4. The data is available (also publicly) for further analysis, if a more specific request can be provided.
- For Figure S4, please explain why the phylogenetic tree has two major branches.
This tree describes the diversity of samples which came through our laboratory from individual patients between January 2014 and December 2022 and fourteen were obtained from the Institute Pasteur’s strain collection (see under Materials and methos). As such, it may not reflect the whole species diversity and should not be discussed as such. Similar patterns have indeed been seen previously, and this has been added to the text (lines 220-221).
- Line 175-177, please clarify the statement by providing data.
The statement was better formulated (lines 223-226).
- Line 2-37 in page 10 is the repeat of line 184-219. Please check the error.
Done, thanks for the correction.
- Line 41, the authors will need to perform additional tests to confirm the existence of subpopulations.
To address this issue, the term ‘putative’ was added before ‘resistant subpopulations’ (line 44).
Reviewer 2 Report
1. Authors need to explain the purpose of the study
2. Authors failed to mentioned about the confirmation of Acinetobacter baumannii clinical isolates. Please explain briefly.
3. Have authors took ethical or consent form?
4. The title of the manuscript is too lengthy and needs to be revised.
5. Introduction part needs to be revised and add information about studied bacteria and its significance regarding your country and then worldwide also.
Author Response
- Authors need to explain the purpose of the study
The aim of the study was reformulated and better summarized in the ‘Introduction’ (lines 81-88).
- Authors failed to mention about the confirmation of Acinetobacter baumannii clinical isolates. Please explain briefly.
In this regard a little paragraph was added in the ‘Materials and Methods’ section (lines 98-102).
- Have authors took ethical or consent form?
There was no need for ethical consent when using biological materials derived from anonymized patients.
- The title of the manuscript is too lengthy and needs to be revised.
The title has been shortened.
- Introduction part needs to be revised and add information about studied bacteria and its significance regarding your country and then worldwide also.
The introduction has been thoroughly revised. The importance to have an accurate test to detect cefiderocol resistance in CRAB (and synergism with avibactam in cefiderocol-resistant-CRAB) has been emphasized also in consideration of the higher prevalence of cefiderocol resistance among CRAB compared to carbapenem-susceptible A. baumannii isolates (lines 73-77, 81-82). Also, the value of the studied bacteria has been highlighted in lines 86-88.
Reviewer 3 Report
First, abbreviations of names and then extensions, e.g. line 33 or 39 Latin names of strains are written in italics, e.g. line 186.
Author Response
First, abbreviations of names and then extensions, e.g. line 33 or 39 Latin names of strains are written in italics, e.g. line 186.
The text was checked thoroughly and corrected.

Round 2
Reviewer 1 Report
I have no further comments.
Too many short paragraphs which can be combined.
Reviewer 2 Report
The authors did a great job and revised manuscript is now much improved. Therefore, I recommend publication.